# Clinical variables associated with immune checkpoint inhibitor outcomes in patients with metastatic urothelial carcinoma: a multicentre retrospective cohort study

Soumaya Labidi,[1,2] Nicholas Meti,[2,3] Reeta Barua,[4] Mengqi Li,[5,6] Jamila Riromar,[7] Di Maria Jiang,[8] Nazanin Fallah-Rad,[8] Srikala S Sridhar,[8] Sonia V Del Rincon,[2,5,6] Rossanna C Pezo,[9] Cristiano Ferrario,[1,2] Susanna Cheng,[9] Adrian G Sacher,[8] April A N Rose [iD] [1,2,5,6]

For numbered affiliations see end of article.

**Correspondence to**
Dr April A N Rose;
april.rose@mcgill.ca

## ABSTRACT

**Objectives** Immune checkpoint inhibitors (ICIs) are indicated for metastatic urothelial cancer (mUC), but predictive and prognostic factors are lacking. We investigated clinical variables associated with ICI outcomes.

**Methods** We performed a multicentre retrospective cohort study of 135 patients who received ICI for mUC, 2016–2021, at three Canadian centres. Clinical characteristics, body mass index (BMI), metastatic sites, neutrophil-to-lymphocyte ratio (NLR), response and survival were abstracted from chart review.

**Results** We identified 135 patients and 62% had received ICI as a second-line or later treatment for mUC. A BMI ≥25 was significantly correlated to a higher overall response rate (ORR) (45.4% vs 16.3%, p value=0.020). Patients with BMI ≥30 experienced longer median overall survival (OS) of 24.8 vs 14.4 for 25≤BMI<30 and 8.5 months for BMI <25 (p value=0.012). The ORR was lower in the presence of bone metastases (16% vs 41%, p value=0.006) and liver metastases (16% vs 39%, p value=0.013). Metastatic lymph nodes were correlated with higher ORR (40% vs 20%, p value=0.032). The median OS for bone metastases was 7.3 versus 18 months (p value <0.001). Patients with liver metastases had a median OS of 8.6 versus 15 months (p value=0.006). No difference for lymph nodes metastases (13.5 vs 12.7 months, p value=0.175) was found. NLR ≥4 had worse OS (8.2 vs 17.7 months, p value=0.0001). In multivariate analysis, BMI ≥30, bone metastases, NLR ≥4, performance status ≥2 and line of ICI ≥2 were independent factors for OS.

**Conclusions** Our data identified BMI and bone metastases as novel clinical biomarkers that were independently associated with ICI outcomes in mUC. External and prospective validation are warranted.

## INTRODUCTION

Bladder cancer represents the 10th most common cancer in the world, with approximately 550 000 new cases annually, and

### STRENGTHS AND LIMITATIONS OF THIS STUDY

⇒ This is a multicentre cohort study, with a large number of patients, treated with immune checkpoint inhibitors for metastatic urothelial cancer at three Canadian institutions.
⇒ We identified reproducible clinical prognostic factors related with response and survival outcomes, including bone metastases for which there are limited data available from prospective studies.
⇒ The retrospective study design may lead to selection bias related to population studied.

accounts for 2.1% of all cancer deaths according to GLOBOCAN 2020.[1] There is a male predominance and smoking is the most common risk factor. In most cases, urothelial carcinomas arise from the urothelium of the bladder but can also develop from the upper urinary tract (renal pelvis and ureters). Tumours invading the detrusor muscle, that is, muscle invasive tumours, account with upfront metastatic disease in 25% of the cases.[2] Despite multimodality management for non-metastatic muscle invasive bladder cancer, 50% of these patients will relapse, with distant metastases in most of the cases.[3 4] Platinum-based chemotherapy remains the standard first-line treatment for metastatic disease.[3 4] Patients who respond or have stable disease following chemotherapy are eligible for subsequent maintenance therapy with the anti-programmed death ligand 1 (PD-L1) immune checkpoint inhibitor (ICI), Avelumab,[5] while those who fail to respond to chemotherapy subsequently can receive the anti-programmed death 1 (PD-1) ICI, Pembrolizumab.[6] Other treatment

options in the platinum-refractory setting include Enfortumab Vedotin and Erdafitinib in patients with susceptible Fibroblast Growth Factor Receptor alterations, and chemotherapy with taxanes (Paclitaxel and Docetaxel) or Vinflunine.[7–10] In the KEYNOTE-045 randomised phase III trial, Pembrolizumab showed a significantly longer overall survival (OS) of 10.3 months versus 7.4 months in patients with platinum refractory metastatic urothelial cancer (mUC) compared with chemotherapy (Docetaxel, Paclitaxel or Vinflunine) (HR: 0.73, 95% CI: 0.59 to 0.91, p value=0.002), with a lower rate of any grade adverse events (60.9% vs 90.2%).[6] Long-term results confirmed an OS benefit and a favourable safety profile.[11] The PD-L1 checkpoint inhibitor Atezolizumab was also associated with durable objective responses in the IMvigpr210 phase II trial, with favourable safety profile,[12] but failed to show survival benefit over chemotherapy in the IMvigor211 phase III trial.[13] In phase I/II trials CheckMate275 and CheckMate032, Nivolumab, a PD-1 inhibitor, achieved objective responses alone or in combination with Ipilimumab, with a manageable adverse events profile.[14 15] Despite responses seen with ICI in the platinum-refractory setting, not all patients will derive benefit and there is a lack of valid prognostic and predictive biomarkers.

In this study, we aimed to evaluate if clinical prognostic markers, that have been defined for other cancer types, are relevant to mUC. In fact, several retrospective studies have addressed the relationship between obesity and outcomes in patients receiving ICI. In melanoma, non-small cell lung cancer (NSCLC) and renal cell cancer (RCC), obesity appears to be linked to better progression-free survival (PFS) and OS.[16–18] And a recent systematic review of 18 retrospective studies across different cancer types, by Indini *et al*, also found an association between high body mass index (BMI) and improved ICI outcomes; yet a strong positive correlation could not be concluded due to the heterogeneity of the studies.[19] As such, the relationship between BMI and immunotherapy outcomes in urothelial cancer remains poorly understood.

The tumour immune microenvironment of different metastatic locations may also affect the response to ICI.[20 21] Retrospective studies reported differences in organ-specific responses and organ-specific OS with immunotherapy in several types of cancer.[22–24]

Neutrophil-to-lymphocyte ratio (NLR) is an available marker of systemic response to inflammation, derived from the absolute counts of neutrophils and lymphocytes on a blood count.[25] It is a well-established poor prognostic factor in several cancers, independently of treatment type.[25–27]

The aim of this retrospective analysis was to assess the association between high BMI, site of metastases, NLR and outcome in terms of response and OS in a population of patients with mUC treated with ICI.

## Patients and methods

### Patient population, characteristics and outcome

We performed a multicentre retrospective cohort study analysis of patients with mUC, from three Canadian cancer centres: Segal Cancer Center, Jewish General Hospital (JGH), Princess Margaret Cancer Center, University Health Network (PM-UHN) and Odette Cancer Center, Sunnybrook Health Sciences Center (SHSC). Patients with histologically proven mUC, who received at least one dose of anti-PD-1/anti-PD-L1 ICI, between December 2016 and January 2021, were included regardless of gender, age and performance status. ICI at any line of treatment for metastatic disease, alone or in combination with chemotherapy or another ICI, was allowed. Patients' characteristics were abstracted from chart review. The following clinical characteristics were collected: age, gender, smoking status, comorbidities, primary tumour location and histology, type and line of ICI treatment, number and type of previous treatment lines received and sites of metastases. BMI at diagnosis, prior to ICI treatment and at progression was assessed. We selected three groups according to BMI at start of ICI: BMI <25, 25≤BMI<30 and BMI ≥30. We collected the following outcomes criteria: ICI overall response rate (ORR) and OS. ICI ORR was determined by investigator assessment of radiological response as per Response Evaluation Criteria in Solid Tumours (RECIST) and was the sum of complete response (CR) and partial response (PR). OS was calculated from the start of ICI treatment to death or last follow-up.

### Statistical analysis

Fisher's exact test and $\chi^2$ test were used to assess differences in response rates between the predefined groups. OS was assessed using Kaplan-Meier method. Log-rank was used to compare groups and Cox regression models were used to perform univariable and multivariable analysis. P value <0.05 was considered statistically significant. Clinical variables that were associated with a p value <0.05 in the univariate analysis were included in the multivariable model. The final multivariable model included all variables with a p value <0.05 in the multivariate analysis. The survival analysis was performed with IBM SPSS V.23 (IBM Corp, Armonk, New York, USA). Multivariable analyses were performed with Stata Statistical Software/MP Release V.17, College Station, Texas: StataCorp LLC. The figures were created using the GraphPad Prism V.10.1.2 for Windows, GraphPad Software ( www.graphpad.com).

### Patient and public involvement

Patients and/or the public were not involved in the design, conduct, reporting and dissemination plans of this research.

**Table 1** Patients' characteristics

| Clinical characteristics | N (%) |
|---|---|
| Age | |
| ≤60 years | 28 (20.7%) |
| >60 years | 107 (79.3%) |
| Gender | |
| Male | 101 (74.8%) |
| Female | 34 (25.2%) |
| Smoking status | |
| Ever smoker | 69 (51.1%) |
| Never smoker | 60 (44.4%) |
| Missing data | 6 (4.5%) |
| Primary tumour site | |
| Bladder | 125 (92.6%) |
| Upper tract | 10 (7.4%) |
| Pathology | |
| Urothelial | 123 (91.1%) |
| Squamous | 7 (5.2%) |
| Other | 5 (3.7%) |
| ECOG | |
| 0–1 | 112 (83.0%) |
| ≥2 | 15 (11.1%) |
| Missing data | 8 (5.9%) |
| Metastatic sites* | |
| LN | 90 (66.7%) |
| Lung | 48 (35.6%) |
| Bone | 43 (31.9%) |
| Liver | 37 (27.4%) |
| BMI | |
| <25 | 55 (40.7%) |
| 25–30 | 43 (31.9%) |
| ≥30 | 23 (17.0%) |
| Missing data | 14 (10.4%) |
| ICI line | |
| First | 51 (37.8%) |
| Second or later | 84 (62.2%) |

*Numbers do not add up to 100% because many patients had multiple sites of metastases.
BMI, body mass index; ECOG, Eastern Cooperative Oncology Group Performance Status Scale; ICI, immune checkpoint inhibitor; LN, lymph node.

## RESULTS

### Patients' characteristics

We identified 135 patients, who received at least one dose of ICI for mUC. The median age was 70 years (26–91 years). Most of the patients had a primary bladder cancer (n=125, 92.6%). Most patients (n=84, 62%) received ICI as a second-line or later treatment for mUC. The median follow-up period was 14.5 months. Patients' characteristics for the entire cohort are shown in table 1.

BMI data were available for 121 patients. A BMI ≥25 was observed in 48.8% of the patients. Median BMI was 20.9, 26.6 and 34.6 in groups BMI <25, 25≤BMI<30 and BMI ≥30, respectively. At the time ICI treatment was initiated, metastatic sites were found as follows: lymph nodes 66.7% (n=90), lung 35.6% (n=48), bones 31.9% (n=43) and liver 27.4% (n=37).

### The relationship between BMI and ICI response and survival outcomes

We observed differences in ORR according to the BMI category of the patient. The ORR was 45.4% in the BMI ≥25 group, versus 16.3% in the BMI <25 group (p value=0.020) (figure 1A). In the BMI ≥25 and BMI <25 groups, we observed a higher proportion of CR to ICI (8 CR vs 1, 22 PR vs 8 and 7 SD vs 10) (online supplemental table 1). Patients with BMI ≥30 experienced the longest median OS (24.8 months) compared with those with 25≤BMI<30 (14.4 months) and BMI<25 (8.5 months) (p value=0.012). (figure 1B). BMI ≥30 remained an independent prognostic variable in multivariable analysis (HR=0.40, 95% CI: 0.17 to 0.96, p value=0.040) (table 2). These results were confirmed for patients treated with ICI alone (online supplemental figure 3A).

### Metastatic sites' response and survival outcomes

The ORR for the entire cohort was 32.2%. The ORR was significantly lower in patients with bone metastases (16% vs 41%, p=0.006) (figure 2A, online supplemental figure 3B), or liver metastases (16% vs 39%, p value=0.013) (figure 2B). Conversely, the presence of metastatic disease involving the lymph nodes was significantly correlated with higher ORR (40% vs 20%, p value=0.032) (figure 2C). The difference in ORR for lung metastases was not statistically significant (27% vs 36%, p value=0.340) (figure 2D). Detailed responses are listed in online supplemental table 1. The median OS was evaluated for the entire cohort, for each metastatic site and for each BMI group. In the entire cohort, median OS was 12.7 months. Bone, liver and lung metastases correlated with significantly shorter survival. The median OS for patients with bone metastases was 7.3 months compared with 18 months in the absence of bone metastases (p value <0.001) (figure 3A, online supplemental figure 3B). Patients with liver metastases had a median OS of 8.6 months compared with 15 months (p value=0.006) (figure 3B), and 8.7 months compared with 17.3 months for those with lung metastases (p value=0.004) (figure 3C). The presence of lymph node metastases was not significantly associated with OS of 13.5 months versus 12.7 months (p value=0.175) (figure 3D). Although bone, lung and liver metastases were all associated with significantly worse OS in univariable analyses, bone metastases were the only metastatic site that remained significantly associated with shorter OS in multivariable analysis (HR=1.98, 95% CI: 1.17 to 3.35, p value=0.010) (table 2).

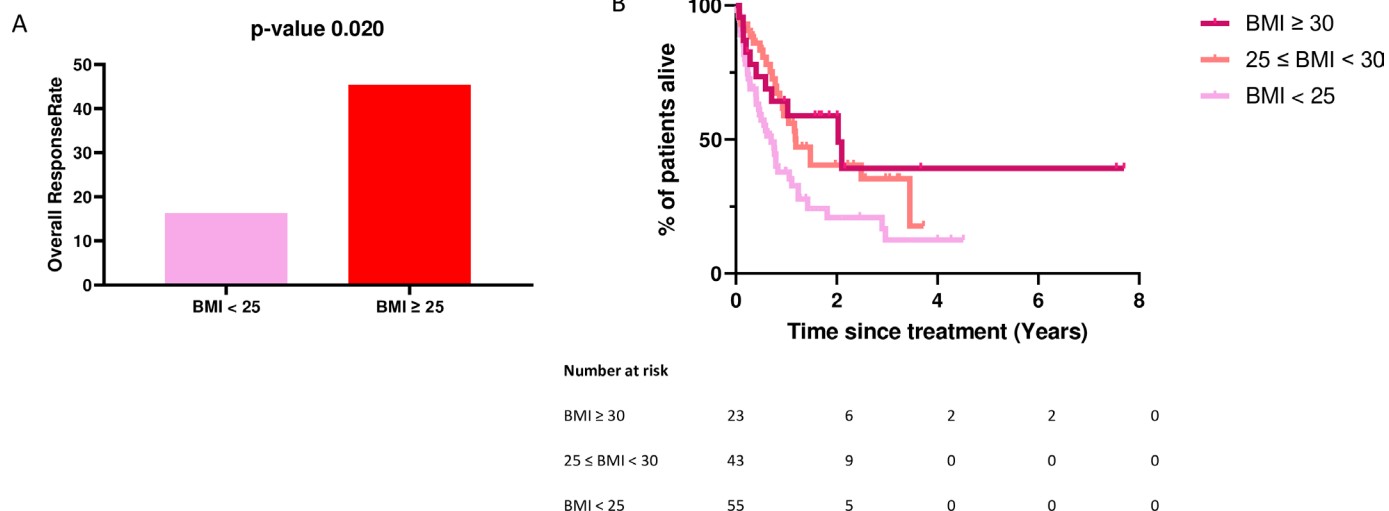

**Figure 1** Overall response rate (A) and overall survival (B) according to BMI. BMI, body mass index.

### NLR response and survival outcomes

Next, we evaluated the relationship between the NLR and clinical outcomes with ICI in this cohort. The ORR was not statistically different for patients with NLR ≥4 compared with those with NLR <4, respectively 24.5% and 36.9% (p value=0.141) (online supplemental figures 1 and 3C). However, NLR <4 was correlated with better OS, with a median of 17.7 months versus 8.2 months for NLR ≥4 (p value <0.001) (online supplemental figures 2 and 3C). The survival benefit associated with a low NLR at the time of initiating ICI therapy was independent of other prognostic variable in multivariable analysis (table 2). We assessed whether there was a relationship between NLR and BMI, but we did not observe any significant correlation (data not shown).

### DISCUSSION

ICIs are approved for a wide array of cancers based on tumour type (melanoma, NSCLC, urothelial cancer)

and/or in a tumour type agnostic manner based on the presence of molecular biomarkers such as high tumour mutation burden[28–30] and microsatellite instability.[30–32] However, in most cases, many patients who are eligible for immunotherapies based on tumour type or biomarker status will either not respond to treatment or will eventually become resistant to treatment. In this analysis, we identified several important clinical biomarkers that associate with better (high BMI) or worse (bone metastases, high NLR) outcomes with ICI in patients with mUC. These biomarkers are easily assessed without any additional costly molecular testing. Moreover, ICIs are expensive, potentially toxic and in mUC only 30% of the patients benefit from ICIs. As such, we believe that these data may be clinically useful to guide treatment decisions.

According to the WHO obesity, classified as a BMI greater than $30 \, \text{kg/m}^2$, is a rising epidemic.[33] Our data showed strong correlation between elevated BMI and improved outcome in terms of response as well as

---

**Table 2** Univariable and multivariable analysis for OS

| OS | Univariable | | | Multivariable | | |
|---|---|---|---|---|---|---|
| | HR | 95% CI | P value | HR | 95% CI | P value |
| BMI ≥30 vs <30 | 0.63 | 0.45 to 0.88 | 0.012 | 0.40 | 0.17 to 0.96 | 0.040 |
| Male vs female | 0.95 | 0.58 to 1.57 | 0.867 | | | |
| Age ≥60 vs <60 | 1.88 | 0.65 to 5.45 | 0.235 | | | |
| ECOG ≥2 vs 0–1 | 1.68 | 0.86 to 3.29 | 0.125 | 2.21 | 1.02 to 4.78 | 0.042 |
| ICI line ≥2 vs 1 | 2.39 | 1.43 to 3.97 | 0.001 | 1.80 | 1.31 to 2.48 | 0.000 |
| NLR ≥4 vs <4 | 2.46 | 1.53 to 3.94 | 0.000 | 2.66 | 1.58 to 4.49 | 0.000 |
| Bone metastasis present vs absent | 2.38 | 1.52 to 3.73 | 0.000 | 1.98 | 1.17 to 3.35 | 0.010 |
| Lung metastasis present vs absent | 1.88 | 1.21 to 2.93 | 0.005 | | | |
| Liver metastasis present vs absent | 1.92 | 1.2 to 3.09 | 0.006 | | | |

BMI, body mass index; ECOG, Eastern Cooperative Oncology Group Performance Status Scale; ICI, immune checkpoint inhibitor; NLR, neutrophil-to-lymphocyte ratio; OS, overall survival.

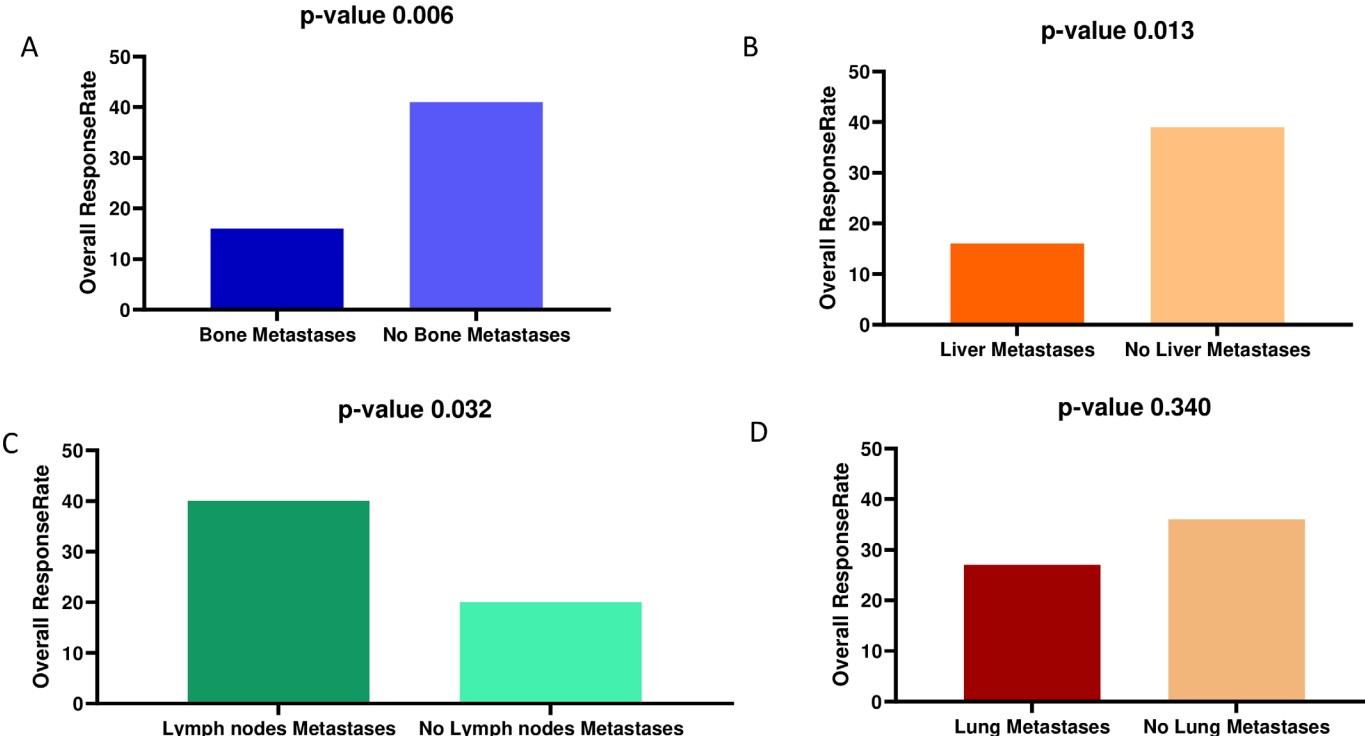

**Figure 2** Overall response rate according to metastatic sites: (A) bone, (B) liver, (C) lymph nodes and (D) lung.

survival. To the best of our knowledge, the present study is the largest to report data of ICI outcomes correlation with BMI in mUC.[19 34 35] A favourable prognostic role of high BMI was also reported in RCC, NSCLC and melanoma.[17 34 36–38] A pooled post-hoc analysis of individual participant data from four prospective trials with Atezolizumab in metastatic NSCLC showed significant

difference in survival between normal weight, overweight and obese patients with improved OS for patients with obesity and overweight compared with normal weight.[17] In a retrospective multicohort analysis of 1918 patients with metastatic melanoma treated with chemotherapy, targeted therapies and ICI, McQuade *et al* studied association between BMI and outcome, as well as interactions

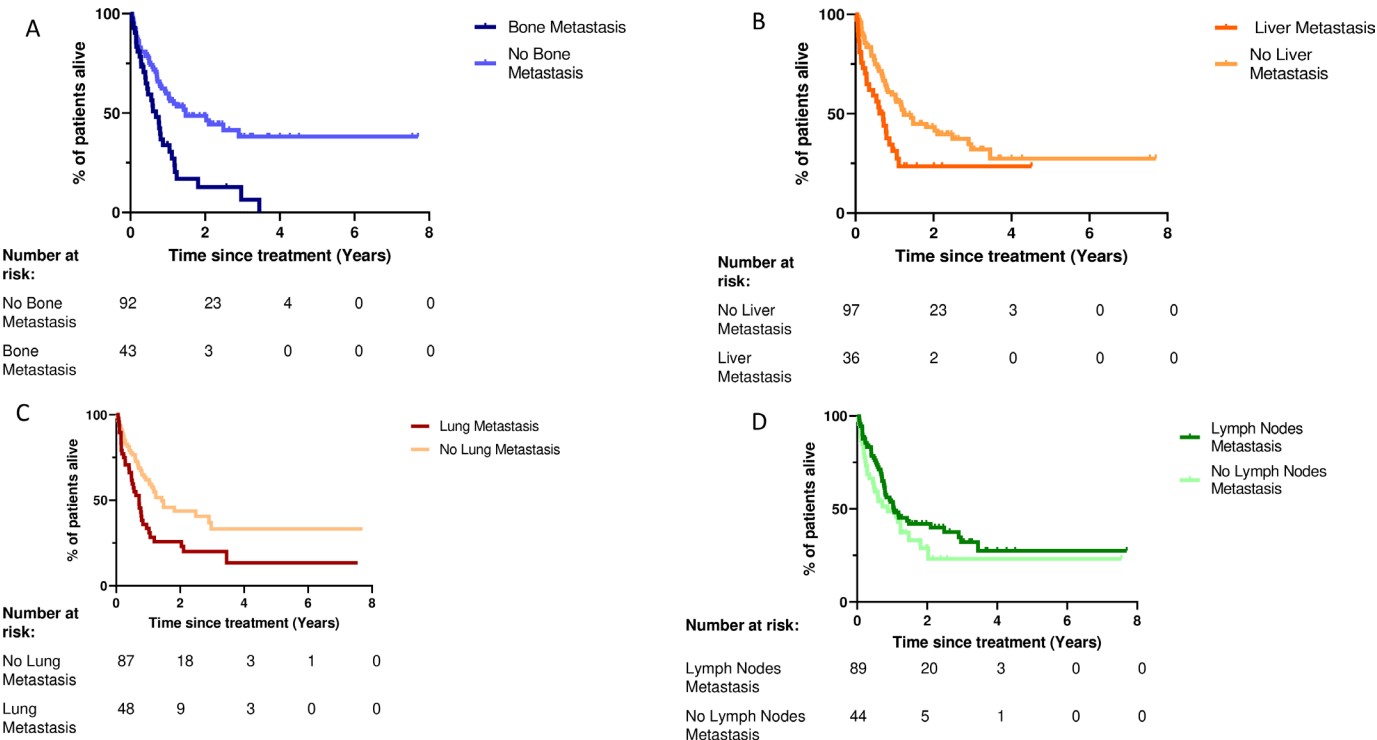

**Figure 3** Overall survival according to metastatic sites: (A) bone, (B) liver, (C) lung and (D) lymph nodes.

between BMI and gender and type of treatment. Obesity was associated with improved OS and PFS with a benefit restricted to patients treated with targeted therapies and ICI.[16]

There is increasing evidence that excess body weight is a modifiable risk factor for several malignancies, including pancreas, kidney, colorectal, postmenopausal breast, ovarian, gallbladder and thyroid cancers.[39] However, obesity is not an established risk factor for bladder cancer and the impact of obesity on cancer survival is less clear. Some studies have reported that obesity is associated with improved survival in some, but not all, patients with cancer, which is referred to the 'obesity paradox'.[40] Numerous studies have shown that obesity can cause immune cell dysfunction,[41–44] while others argue that the chronic low-grade inflammation underpinning obesity creates a proinflammatory state that synergises with immunotherapy.[45 46]

Preclinical and translational research has provided insight into how immune checkpoint blockade can benefit obese patients. Obesity appears to influence T cell function and phenotype. For example, leptin-associated T cell dysfunction due to obesity has been observed across different species and tumour models and was found to enhance the immune response to anti-PD-1 therapy.[41] In a mouse study, Shirakawa *et al* reported that obesity caused a preferential increase and accumulation of senescent CD44$^{hi}$CD62L$^{lo}$CD4+ T cells that constitutively express PD-1 and CD153.[47] Moreover, a study in a colon carcinoma mouse model showed that obesity impaired the infiltration, cytokine production and metabolic activity of tumor-infiltrating CD8 T cells, while anti-PD-1 restored CD8 T cells infiltration, proinflammatory cytokine production and metabolic activity, leading to complete tumour eradication and immune memory.[48] Obesity may also affect other immune cells in the tumour microenvironment (TME), such as myeloid-derived suppressor cells (MDSCs), which are known to suppress T cell responses. MDSCs can suppress T cell activity through many mechanisms, one of which is expressing PD-L1 to induce T cell exhaustion.[49 50] In a preclinical study, inhibiting PD-L1 on MDSCs decreased their ability to suppress T cell activity, suggesting that obesity could also accelerate tumour progression by promoting the expression of PD-L1 on MDSCs.[51] Finally, a preclinical study in a breast cancer model showed that although obesity accelerated tumour progression, anti-PD-1 treatment significantly reduced tumour burden by reshaping the local TME landscape.[52]

The relationship between obesity and ICI is complex and depends on various of factors, including the types of cancer and ICI involved. However, BMI may be an imperfect marker and its evaluation based only on weight prior to ICI start can be limiting. Assessment of the weight loss or change over the time may be a better indicator of disease prognosis. As BMI cannot distinguish body fat from muscle, it may not be the best tool to assess obesity.[53] Other measurement methods could be used for better assessment such as dual energy X-ray absorptiometry or fat referenced quantitative MRI.[54]

Further studies are needed to understand the molecular mechanisms underlying the interaction between obesity and cancer immunity and to identify potential targets for effective interventions.

Bone metastases are a validated negative prognostic factor in mUC treated with platinum-based chemotherapy.[55 56] In our study, poor response and survival outcome of bone and liver metastases was significant, while lymph nodes metastases correlated with higher ORR. These results are consistent with published data in the literature for patients treated with ICI.[57–59] Makrakis *et al* showed lower response rates and shorter OS for bone and liver metastases on retrospective data from 917 mUC treated with ICI as first or ≥ second line, but higher ORR for lymph node-confined metastases.[57] A retrospective multicentric Japanese study reported shorter ORR for bone metastases in a cohort of patients treated with Pembrolizumab for mUC.[58] Similar results have also been reported from prospective trials.[60 61] In the IMvigor210 phase II trial, the ORR on Atezolizumab was 32% for lymph node-confined metastases, and only 8% for liver metastases, data for bone metastases were not reported.[60] Higher ORR for lymph nodes metastases (47%) compared with liver metastases (23%) was also reported in the KEYNOTE-052 trial.[61] Prognostic models for patients with mUC treated with ICI identified liver or visceral metastases as poor prognostic factors.[62 63] Sonpavde *et al* developed a five-factor prognostic model for survival for mUC treated with Atezolizumab, Avelumab or Durvalumab, based on the data from phase I and II trials.[64] They identified three risk groups based on performance status, liver metastases, platelet count, NLR and lactate dehydrogenase (LDH) levels as prognostic factors.[64] The 1-year Kaplan-Meier survival estimates of those in the low, intermediate and poor risk groups were 76.2%, 33.8% and 8.6%, respectively.[64] The authors developed an interactive web-based tool to calculate the expected survival probabilities according to the five factors.[64] In our study, LDH levels were not uniformly assessed prior to ICI treatment at all sites, and we did not collect data for platelet count. Therefore, we could not compare the prognostic utility of this tool in our cohort of patients. Most prospective trials of ICI in mUC do not use presence/absence of bone metastases as a stratification factor, despite the fact that bone metastases are associated with poor response to chemotherapy and impact quality of life.[55 56 65] Owari *et al* validated a specific prognostic scoring system, B-FOM, to predict survival for patients with bone metastasis from different genitourinary cancers based on five prognostic factors: primary tumour (prostate, renal or urothelial cancers), poor performance status, visceral metastases, high Glasgow-prognostic score and elevated NLR.[66] This prediction tool may be helpful to individualise optimal treatment strategy for the patients. A better understanding of the bone microenvironment is crucial to help the development of novel therapeutic strategies and improve outcomes.[67]

The distant organ microenvironment, also known as metastatic microenvironment (MME), plays an important role in site-specific metastases.[68] For instance, macrophages, which are abundant immune cells in the lungs, can bind to cancer cells via receptor VCAM-1 transmit, signalling a chain of events leading to lung-specific metastases in breast cancer.[69] It has been demonstrated that myeloid cells can remodel the premetastatic lung from an immune protective state to a state favouring tumour progression, thereby promoting lung metastases.[70] Furthermore, lung stromal cells can also promote tumour colonisation and metastasis by secreting periostin.[71] In the liver, there is a significant presence of circulating and resident natural killer (NK) cells, which serve as the primary effectors of liver immune function. This NK cell maintained the breast cancer cells dormant in the liver by secreting interferon-γ (IFN-γ), while sustaining NK cell abundance with interleukin-15 (IL-15)-based immunotherapy succeeded to prevent liver metastases and prolong the survival in preclinical models.[72] Conversely, bone forms an immunosuppressive environment mainly due to immature NK cells, a small number of cytotoxic T cells, a large number of myeloid progenitors and Treg.[73] In a human bladder tumour xenografts model, researchers observed a high infiltration of bone marrow-derived host CD11b myeloid cells. They further demonstrated that enhanced tumour-associated in the TME promote an immunosuppressive protumoral myeloid phenotype.[74] These observations demonstrate how the MME can influence the metastatic potential and outcome of different types of cancer cells in different organs, suggesting that MME might also potentiate site-specific metastases in bladder cancer. Taken together, all these studies suggest the importance of understanding the metastatic potential and outcome of different types of cancer cells in different organs, through which we can further develop more effective therapeutic measures against bladder cancer.

NLR is a well-established poor prognostic factor in several cancers, independently of treatment type.[25–27] It could be considered as a surrogate marker of chronic inflammation and evasion of immune surveillance.[75] In our study, high NLR was associated with shorter median OS. A recent study by Valero *et al* showed poor response and survival outcomes, for 1917 patients with high NLR at diagnosis, treated with ICI for multiple cancer types.[75] Similar results were reported for melanoma and NSCLC.[26 76] Banna *et al* explored the prognostic and predictive role of NLR and LDH in mUCs treated with ICI and showed significant correlation with PFS and OS for high NLR.[77] For patients treated with ICI for mUC, a high NLR prior to first-line chemotherapy and second-line pembrolizumab was associated with worse survival outcomes.[78] NLR may represent an accessible low-cost predictive biomarker.

The present study has some strengths and limitations. It presents a relatively large multicentre cohort of patients with significant prognostic factors. On the other hand, this is a retrospective study, with heterogenous population and an investigator-based response assessment on clinical and radiological chart review. However, the observations made in this analysis are hypothesis generating. The evaluation of obesity and its relationship with ICI response warrants future prospective studies that incorporate better established measures of obesity and metabolic syndrome. Moreover, little is known about how bladder cancer cells interact with the liver and bone TME and this warrants further investigation in preclinical models to develop optimal immunotherapies for patients with bone and liver mUC.

## CONCLUSION

Our study identified elevated BMI, NLR and presence of bone metastases as potential biomarkers for ICI response and survival in mUC. Obesity was associated with improved survival and response rate, whereas bone metastases and high NLR are associated with lack of response and shorter survival. Prospective validation of these data is warranted, especially in the evolving landscape of therapeutic options for mUC with novel agents and combinations with ICI.[79]

**Author affiliations**
[1]Segal Cancer Centre, Jewish General Hospital, Montreal, Québec, Canada
[2]Gerald Bronfman Department of Oncology, McGill University, Montreal, Québec, Canada
[3]St Mary Hospital, Montreal, Quebec, Canada
[4]Toronto East Health Network Michael Garron Hospital, Toronto, Ontario, Canada
[5]Lady Davis Institute for Medical Research, Montreal, Québec, Canada
[6]Division of Experimental Medicine, Faculty of Medicine, McGill University, Montreal, Quebec, Canada
[7]National Oncology Center, The Royal Hospital, Seeb, Muscat, Oman
[8]Medical Oncology, Princess Margaret Hospital Cancer Centre, Toronto, Ontario, Canada
[9]Odette Cancer Center, Sunnybrook Health Sciences Center, Toronto, Ontario, Canada

**Contributors** Guarantor: AANR. Guarantor of integrity of the entire study: AANR, SL. Study concepts and design: AGS, AANR. Literature research: SL, ML. Data collection: SL, NM, RB, JR, AANR. Statistical analysis: SL, AANR. Manuscript preparation: SL, ML, AANR. Manuscript editing review and critical analysis: SL, NM, RB, ML, JR, DMJ, NF-R, SSS, SVDR, RCP, CF, SC, AGS, AANR. All authors were involved in data interpretation, critical appraisal of the draft manuscript and gave final approval on the submitted version.

**Funding** April A N Rose acknowledges support from a Fonds de Recherche du Québec – Santé (FRQS) Clinical Research Scholar Award. This research was supported by a University of Toronto, Medical Oncology Strategic Planning Medical Oncology Innovation Fund to Adrian G Sacher and April A N Rose and funding from the Jewish General Hospital (JGH) Foundation to April A N Rose.

**Competing interests** Nicholas Meti—Honoraria: Takeda, Novartis. Consulting: Takeda, Seagen, AstraZeneca, Pfizer. Srikala S Sridhar—Consulting: Astellas, AstraZeneca, Bayer, Bicycle Therapeutics, BMS, Eisai, EMD Serono, Gilead, Ipsen, Janssen, Merck, Pfizer, Seagen; Research funding (Institution): Janssen, Seagen, EMD Serono. Rossanna C Pezo—Honoraria: Pfizer, Novartis, BMS, Merck, Sanofi, Gilead, AstraZeneca; Research funding (Institution): Merck, Novartis, Pfizer, Taiho, Jazz Pharmaceuticals, ZymeWorks, AstraZeneca. Cristiano Ferrario—Honoraria: Pfizer, Bayer, Novartis, Roche, Bayer, Knight therapeutics, AstraZeneca, Merck, Astellas Pharma; Consulting—Merck, Novartis, AstraZeneca, Roche; Research funding (Institution): Merck, Novartis, Pfizer, Lilly, ZymeWorks, AstraZeneca, Janssen Oncology, Bayer, Astellas Pharma, Roche, Sanofi, Seattle Genetics, Semonix Pharmaceuticals, Bicycle Therapeutics, Immunomedics. Susanna

Cheng—Consulting: AstraZeneca, Merck. April A N Rose—Consulting: EMD Serono, Advanced accelerator applications/Novartis; Research funding (Personal): Canadian Institutes of Health Research CIHR, Canadian Cancer Society, Conquer Cancer Foundation, Jewish General Hospital Foundation, TransMedTech Institute, Canada Foundation of Innovation; Research Funding (Institution): AstraZeneca, Merck, Pfizer, Seattle Genetics; Employment: Family member Merck. Soumaya Labidi, Mengqi Li, Sonia V Del Rincon, Reeta Barua, Jamila Riromar, Di Maria Jiang, Nazanin Fallah-Rad, Adrian G Sacher: no competing interests.

**Patient and public involvement** Patients and/or the public were not involved in the design, or conduct, or reporting or dissemination plans of this research.

**Patient consent for publication** Not applicable.

**Ethics approval** Ethics approval for this study was obtained from the Clinical Trials Ontario (CTO Project ID: 2067) after review by the University Health Network Research Ethics Board on 19 August 2020, for PM-UHN and SHSC sites. Ethics approval for the JGH site was approved by theIntegrated University Health And Social Services Centres (CIUSSS) West Central Montreal REB (Project 2022-2888) on 28 October 2022.

**Provenance and peer review** Not commissioned; externally peer reviewed.

**Data availability statement** All data relevant to the study are included in the article or uploaded as supplementary information.

**ORCID iD**
April A N Rose http://orcid.org/0000-0002-9845-4603

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
