## [Reviewer comments · BMJ Open]

ARTICLE DETAILS

TITLE (PROVISIONAL)	Clinical variables associated with immune checkpoint inhibitor outcomes in patients with metastatic urothelial carcinoma: A multicenter retrospective cohort study
AUTHORS	Rose, April; Labidi, Soumaya; Meti, Nicholas; Barua, Reeta; Li, Mengqi; Riromar, Jamila; Jiang, Di Maria; Fallah-Rad, Nazanin; Sridhar, Srikala; Del Rincon, Sonia V.; Pezo, Rossanna C.; Ferrario, Cristiano; Cheng, Susanna; Sacher, Adrian

VERSION 1 – REVIEW

REVIEWER	Niegisch, Guenter Heinrich Heine University Dusseldorf
REVIEW RETURNED	24-Nov-2023

GENERAL COMMENTS	This is an interesting retrospective analysis regarding prognostic factor. In detail, the authors are able to show that BMI, bone metastases and neutrophils/lymphocyte ratio may be independent prognostic factors regarding treatment response. The manuscript is well written and scientifically sound. However the authors should consider the following remark: - as described in the methods section, the authors included all patients receiving combined IO and chemotherapy treatment. In my opinion, these patients should be excluded from the analysis as it is not possible to relate response and survival either to the chemotherapy or the IO therapy.- The patient characteristics should be briefly summarized in the abstract. Especially the number of patients with first-line and second or later line therapy should be outlined.- Regarding prognostic factors in IO therapy for UC, a 5-factor model based on landmark trials has been published only recently (Sonpavde et al., J Urology 2020). In this analysis NLR was shown to be an independent factor as well. Though developed for 2nd/late line IO therapy, it would be worth to apply this prognostic model to the presented cohort and discuss the results in the context of the authors' findings. For example, in contrast to the 5-factor modeling the present study bone metastases have been identified as prognostic factors while liver metastases have not. Is this due to the characteristics of the patient cohort (too few patients with liver metastases)?- In the multivariate model, ECOG and line of therapy have been identified as independent factors as well. This information is missing in the abstract.
--

REVIEWER	Ferreira, Clara University Hospital of Coventry and Warwickshire, Nuclear Medicine
-----------------	---

REVIEW RETURNED	15-Dec-2023
GENERAL COMMENTS	I have indicated all my comments in the document itself - please check the comments in the .pdf document attached.

VERSION 1 – AUTHOR RESPONSE

Dr. Guenter Niegisch, Heinrich Heine University Dusseldorf

Thank you for all your valuable comments.

-A described in the methods section, the authors included all patient receiving combined IO and chemotherapy treatment. In my opinion, these patients should be excluded from the analysis as it is not possible to relate respond and survival either to the chemotherapy or the IO therapy.

We would like to thank you for this point. We performed a sensitivity analysis excluding the patients who received combined treatment. BMI, bone metastases and NLR were still significantly correlated with overall survival. We included these results as supplemental figures (Supplemental Figure 3).

-The patient characteristic should be briefly summarized in the abstract. Especially the number of patients with first-line and second or later line therapy should be outlined.

-In the multivariate model, ECOG and line of therapy have been identified as independent factors as well. This information is missing in the abstract.

We updated the abstract results section as recommended; however the word count limit does not allow more detailed results reporting in the abstract.

-Regarding prognostic factors in IO therapy for UC, a 5-factor-model based on landmark trials has been published only recently (Sonpavde et al., J Urology 2020). In this analysis NLR was shown to be an independent factor as well. Though developed for 2nd/later line IO therapy, it would be worth to apply this prognostic model to the presented cohort and discuss the results in the context of the authos finding. For example, in contrast to the 5-factor modeling the present study bone metastases have been identified as prognostic factor while liver meets have not. Is this due to the characteristics of the patient cohort (to few patients with liver mets)?

We would like to thank you for this important comment and the reference suggestion. We updated the discussion section and reported the five-factor prognostic model developed by Sonpavde et al. However, in our study, LDH levels were not uniformly assessed prior to ICI treatment at all sites, and we did not collect data for platelet count. Therefore, we could not compare the prognostic utility of this tool in our cohort of patients. In our cohort, only 27.3% of the patients had liver metastases. The presence of liver metastases was correlated with significantly lower response rate and shorter overall survival. But this was not identified in the multivariable analysis, possibly due to the small number of patients with liver metastases.

Miss Clara Ferreira, University Hospital of Coventry and Warwickshire

Thank you for your valuable comments.

First, we would like to report that we addressed all the comments regarding the abbreviations throughout the manuscript, as well as the spelling, orthograph and sentences rewording reviews. We also followed the BMJ open reporting rules for the “p-value” throughout the manuscript as recommended.

-Page 3 of 26, Line 23: Do you mean no significant difference was found in this case? Considering the p-value.

Yes, we mean that there is no statistically significant difference in the overall survival. We clarified that in the text.

-Page 3 of 26, Line 30: What exactly does this mean? In an abstract, the transmitted message to the reader should be clear and I have no idea what this means, especially in the conclusion.

Our study identified clinical markers associated with outcome (response and survival). However, this is a retrospective study, and we recommend that these markers should be validated in a prospective study.

-Page 11 of 26, Line 10: Is this a clinical or pre-clinical study? You cite all types of studies, so please indicate which type of study you are talking in here.

Yes, this is a preclinical study. We updated it as “Finally, a preclinical study in a breast cancer model showed that although obesity accelerated tumor progression, anti-PD-1 treatment significantly reduced tumor burden by reshaping the local TME landscape.”

-Page 11 of 26, Line 25: The reference 54 only refers the methods to study body composition, so which reference do you have to say that it is better to assess the weight loss or change over time?

We added the following reference here: Arnett D.K et al. *2019 ACC/AHA Guideline on the Primary Prevention of Cardiovascular Disease: A Report of the American College of Cardiology/American Heart Association Task Force on Clinical Practice Guidelines*. J Am Coll Cardiol 2019 Vol. 74 Issue 10 Pages e177-e232

-References section

Ref 1: Why are you using a paper from 2018 as a reference? GLOBOCAN released a new paper in 2020. Please check if you can use that new paper as a reference considering that the data is more recent than this one.

Thank you for the reference suggestion. We updated it as follows:

Sung, H., et al., *Global Cancer Statistics 2020: GLOBOCAN Estimates of Incidence and Mortality Worldwide for 36 Cancers in 185 Countries*. CA Cancer J Clin, 2021. **71**(3): p. 209-249.

Ref 3: DOI: 10.1016/j.humpath.2022.08.006 - does this paper have the information that you need? This paper is from June/2023 and it might have more updated information.

Thank you for the suggestion. However, the most recent one is reporting the changes to the classification, and this is not what we are referring to in the text.

Ref 5: Please check the reference "EAU Guidelines. Edn. presented at the EAU Annual Congress Milan 2023. ISBN 978-94-92671-19-6". This is the updated version of the Guideline on Muscle-Invasive and Metastatic Bladder Cancer from EAU.

Thank you for the suggestion. But, in the 2023 version, only updates were added. They are not relevant for our text.

Ref 6: Could you please check the papers with DOI: 10.1016/j.eururo.2023.08.001 and <https://www.nice.org.uk/guidance/ta788>? They might have more updated information.

Thank you for the suggestion. However, this paper reports a subgroup analysis that is not relevant to our text.

Ref 7: Could you please check the paper with DOI: 10.21037/tau.2019.09.19? Does it have the same information that you are looking for? This paper is more recent in comparison to the one that you have.

Thank you for the suggestion. This paper is a commentary on an update of the trials results. Our reference is the first published trial data, and therefore more relevant to our text.

Ref 8: Could you please check the paper with the DOI: 10.1016/j.clgc.2015.09.008? This is a phase II study with a weekly dose of Docetaxel as a Second-Line Chemotherapy in Patients with Metastatic Urothelial Carcinoma. Does it have the information that you want? The paper that you chose to cite is very old, but if it is the right one, please do not change it.

Thank you for the reference suggestion. This article is a phase II trial exploring a different schedule for docetaxel: weekly regimen. Our reference is the first trial with docetaxel in the indication and therefore more relevant to our text.

Ref 11: Have you checked the document in the link below? This is the NICE guideline for the use of Vinflunine for the treatment of advanced or metastatic transitional cell carcinoma of the urothelial tract.

Link: <https://www.nice.org.uk/guidance/ta272/documents/transitional-cell-carcinoma-of-the-urothelial-tract-vinflunine-final-appraisal-determination-document2>

Thank you for the comment. In fact, vinflunine is not recommended by the NICE guidelines, but it is an option in other guidelines such as ESMO for example: DOI:<https://doi.org/10.1016/j.annonc.2021.11.012>

Ref 19: Please check the paper with the DOI: 10.1245/s10434-020-08422-9 - does it have what you need? There are a few papers which are more recent than this one.

Thank you for the suggestion. This reference is an editorial comment on a published paper reporting a retrospective single institution cohort of 2329 patients. Our reference is based on more than 5000 patients multicentric cohort. Therefore, we believe these data are more robust.

Ref 28: Please check the paper with DOI: 10.3892/mco.2020.2095 - it is a much more recent paper. Does it have the information that you need?

Thank you for your reference. The article proposed is discussing NLR in the context of radiation, which is not the context of our study. In fact, our interest is NLR as prognostic factor with immune checkpoints inhibitors treatment.

Ref 34: Please check the paper with the DOI: 10.1016/j.jacadv.2023.100570 - does it have the information that you need? It is an analysis to the document that you cite in here, but a more recent view.

Thank you for the suggestion. However, we favor to keep the published guidelines as a reference.

Ref 41: Please check the papers with the DOI: 10.1002/jcsm.13007 and DOI: 10.1158/0008-5472.CAN-17-3043 - do they have the information that you need? These are more recent papers than the one you cite in here.

Thank you for your suggestions. The first reference (DOI: 10.1002/jcsm.13007) is an editorial. The second reference is assessing the limits of the methodology in the studies of obesity and cancer, which is not what we are reporting in the text.

Ref 48: Please check the paper with DOI: 10.1093/immadv/ltac015 - does it have the information that you need? It is a more recent paper than the one you cite in here.

Thank you for your suggestion, but it does not report the information needed in our text.

Ref 66: Could you take a look at the paper with the DOI: 10.3389/fendo.2022.1019864, please? It is a more recent paper than the one you cite in here.

Thank you for your suggestion. This article is an interesting mini-review, however our reference is a more detailed review and therefore more relevant to the text.

Ref 13, 60 and 61: Thank you for your suggestions. As we report the trials, we favor to refer to the initial publications.

-Tables and Figures section

Thank you for all your valuable comments, which we think helped to make our tables and figures clearer and easier to understand for the readers.

Abbreviations were defined as recommended in all tables, supplemental table and figures.

Table 1: Totals were reviewed and corrected.

Clarification was added to Table 1 regarding metastatic sites: Numbers to not add up to 100% because many patients had metastases to multiple sites.

Table 2: Data were presented from highest to lowest.

“Yes vs No” was corrected to “Present vs Absent.”

Figures: Y axis was defined as “Overall response rate” in all figures as recommended.

Data were presented from lowest to highest as recommended in Figures 1A and Supplemental Figure 1.

Figure 1B and Figures 3 (A, B, C and D): This represents the number of patients at risk to experience an event (death) at the different timepoints. These data are helpful to interpret the survival curves. They are part of the survival curves figures.

In closing, I would like to extend my sincerest gratitude to you and the reviewers for taking the time to review our paper and providing insightful feedback. Based on your comments, we are confident that the revised paper is stronger than the original submission. We hope that you will consider our request for you to re-review this new revised manuscript.

I look forward to hearing from you soon.

Sincerely,

VERSION 2 – REVIEW

REVIEWER	Ferreira, Clara University Hospital of Coventry and Warwickshire, Nuclear Medicine
REVIEW RETURNED	05-Feb-2024

GENERAL COMMENTS	Dear all, Thank you very much for reviewing and discussing about the comments suggested by the editor and the reviewers. The improvement in the paper is huge and I would like to congratulate you for this! I am sure that it means many hours of work, but it was definitely worth it. I would like to specifically congratulate you about the references, the changes in the tables and graphs - they look very good now. I have only some minor changes - could you please review these? Hopefully, after these, the paper should be suitable for publication. Attached you can see the document where I have included all my comments. Kind regards, Clara Ferreira
--

VERSION 2 – AUTHOR RESPONSE

Thank you for your valuable comments. We addressed the changes requested as shown in the file attached.

We are thankful for your comments that made our manuscript stronger.